# University Students’ Mental Health and Well-Being during the COVID-19 Pandemic: Findings from the UniCoVac Qualitative Study

**DOI:** 10.3390/ijerph19159322

**Published:** 2022-07-30

**Authors:** Mayuri Gogoi, Adam Webb, Manish Pareek, Christopher D. Bayliss, Lieve Gies

**Affiliations:** 1Department of Respiratory Sciences, University of Leicester, Leicester LE1 9HN, UK; mg432@le.ac.uk (M.G.); mp426@le.ac.uk (M.P.); 2Department of Genetics and Genome Biology, University of Leicester, Leicester LE1 7RH, UK; ajw51@leicester.ac.uk (A.W.); cdb12@leicester.ac.uk (C.D.B.); 3Department of Infection and HIV Medicine, University Hospitals of Leicester NHS Trust, Leicester LE1 5WW, UK; 4School of Media, Communication and Sociology, University of Leicester, Leicester LE1 7RH, UK

**Keywords:** mental health and well-being, university students, COVID-19 pandemic, isolation, academic and financial concerns, coping and resilience

## Abstract

The worldwide spread of the coronavirus disease 2019 (COVID-19) pandemic in early 2020 affected all major sectors, including higher education. The measures to contain the spread of this deadly disease led to the closure of colleges and universities across the globe, disrupting the lives of millions of students and subjecting them to a new world of online learning. These sudden disturbances coupled with the demands of a new learning system and the experiences of living through a pandemic have placed additional strains on the mental health of university students. Research on university students’ mental health, conducted during the pandemic, have found high levels of stress, anxiety and depression among students. In this qualitative study, we aimed to understand how pandemic experiences have affected student well-being by conducting in-depth interviews with 34 undergraduate students enrolled in a UK university. All interviews were conducted through Microsoft Teams and recorded with prior permission. Transcripts of recorded interviews were thematically analysed which identified two broad themes: (i) University students’ mental health and well-being experiences during the pandemic; (ii) factors that influenced students’ mental health and well-being. These factors were further distributed across six sub-themes: (a) isolation; (b) health and well-being; (c) bereavement; (d) academic concerns; (e) financial worries and; (f) support, coping, and resilience. Our study identifies the importance of mental health support to university students during pandemics and calls for measures to improve access to support services through these crisis points by universities. Findings can also inform students’ mental health and risk assessments in the aftermath of the pandemic.

## 1. Introduction

The world came to a standstill in March 2020 when the World Health Organisation (WHO) declared the coronavirus disease 2019 (COVID-2019) as a pandemic with rising case numbers and mortality rates across the globe [1]. In the United Kingdom (UK), a complete lockdown was announced by the Prime Minister on 23 March 2020 restricting the movement of people outside their homes [2]. As a result, educational institutions, including universities, closed doors and all teaching and assessment moved online. In Autumn 2020, blended learning (in-person and online) was introduced by most UK universities; whilst campus-based teaching is gradually making a come-back, the blended format is likely to continue for some time.

University students are not regarded as a clinically vulnerable group for COVID-19, however, concerns were raised early on about the mental health impacts that the pandemic can have on this group [3,4,5,6]. Research conducted among college and university students in various parts of the world, including the UK, found evidence of increasing levels of anxiety, depression, distress and other mental health conditions among this population during the pandemic [7,8,9,10,11,12,13]. Details of these research and the significant findings have been summarised below in the literature review section.

### Literature Review

The mental health of university students had been a matter of concern even before the pandemic. In the UK, a Parliamentary research briefing published in December 2020 reported a six-fold rise in students’ mental illness since 2010 [14]. Another large-scale survey, carried out in 2019, among students from 140 UK universities found that more than one-quarter (26.6%) of the respondents had received at least one mental health diagnosis in the past, with anxiety and depression being the most commonly diagnosed conditions [15]. In fact, a systematic review carried out more than a decade ago had highlighted the burgeoning problem of mental health among university students and offered recommendations [16]. While individual research has been continuing in this area in the last decade [17], it is the COVID-19 pandemic which brought students mental health back into focus.

Mental health of university students during the pandemic has received considerable attention in research and studies have been reported from across the world on various aspects of mental health among different groups of students [7,8,9,10,11,12,13,18,19]. These studies have reported varying levels of mental health (mainly anxiety, stress and depression) among college and university students during the pandemic. For instance, Ma et al.’s questionnaire study of more than 700,000 Chinese students in early 2020 found that nearly 45% of their cohort had experienced mental health problems, with anxiety being the most commonly experienced symptom [13]. Similarly, another multinational study conducted among university students in nine countries between May–July 2020 found prevalence of high levels of stress (61.3%), depression (40.3%) and anxiety (30%) [20]. Alarming levels of stress and anxiety (71%) were also reported by Wang et al. among students in a Texas university in the United States (US) [8]. Large-scale surveys on mental health of university students conducted in the UK revealed high levels of anxiety and depression, rise in sedentary behaviour and poor sleep quality [9,21,22,23,24]. Chen and Lucock’s study involving 1173 undergraduate and postgraduate students in a UK university found more than 50% of respondents having anxiety and depression levels above clinical cut-off [9].

Mental health problems were reported to be most severe during the early days of the pandemic and lockdown. Carr et al.’s study on mental health trajectories among UK university staff and postgraduate students found anxiety and depression levels to be high during the first lockdown in April 2020 as compared to that in summer 2020 when most lockdown restrictions were lifted in the UK [22]. Another longitudinal survey conducted among 214 UK university students found falling levels of mental well-being and increasing levels of perceived stress during the first lockdown [21]. A similar study conducted in US reported increase in moderate-severe anxiety from 18.1% (in October 2019–February 2020) to 25.3% (in June–July 2020) and increased moderate-severe depression from 21.5% to 31.7% during the same period [7]. Consecutively, mental well-being was also observed to have relatively improved after the first wave of the pandemic with better coping among students [25].

Several factors have been identified as posing a risk or influencing mental health of university students during the pandemic. Most studies have found female students to be at greater risk of poor mental health as compared to males [7,10,12,15,17,18,26]. Ethnicity has also been reported as a risk factor and one study conducted in the US have found Non-Hispanic Black students to be at higher risk of depression [7]. Another US-based study found racial/ethnic discrimination to be significantly associated with high levels of depression [27]. Simultaneously, academic pressure and social isolation were also found to significantly affect students’ mental health with sudden disruption in education, demands of online learning, increased social isolation and a relatively bleak job market acting as new challenges that have arisen during the pandemic [7,8,10,12,24,28]. Students with pre-existing mental health problems were also found to have poorer mental health outcomes during the pandemic [10,12,13,29]. Additionally, lack of social support or low social support have been found to influence students’ mental health negatively during the pandemic [11,13]. Economic stressors such as food insecurity, low economic status and, worsening financial situation have also been observed to affect students’ mental health adversely [11,12,29,30,31]. Some studies have also reported students’ use of media and access to information to have an influence on their mental health, with more exposure to media coverage associated with greater chances of poor mental health outcomes [10,13]. Worries about infection or contracting COVID-19 by themselves or family members was also found to affect students’ mental health [12,13,23], although this was not uniform, and some studies did not find any significant correlation between these variables [7]. Other factors which contributed to the risk or influenced students’ mental health but were not commonly identified include: course/programme of study [10]; year of study [11]; and belief in conspiracy theories [10].

Although most studies have looked at risk factors, some studies have also focussed on coping of students during the pandemic. For instance, a study of 7800 college students in China found resilience, adaptive coping strategies and social support to mediate the relationship between COVID-19 negative experiences and acute stress disorder [32]. An Australian study found students used video chats, social media, took up exercising and engaged in new hobbies to cope with the pandemic [28]. Similar findings have also been presented from a qualitative study conducted among international students registered in an UK university where students shared about watching movies, chatting with friends and family and exercising as coping strategies they employed during the pandemic [33]. Alongside positive coping, some studies also found students engaging in negative coping strategies such as smoking and alcohol consumption to get through the pandemic [9,26,34].

As described in the preceding paragraphs, previous research among students in universities have established varying levels of mental health during the pandemic. While these observational studies have helped in understanding the status of mental health, the findings can be supplemented and enhanced by qualitative studies that offer rich descriptions of students’ self-perspective and lived experience of their own mental health and well-being. We conducted a qualitative study of students to understand their experiences of studying through the pandemic and the impact of these on their mental health and well-being. Our analysis revealed two important considerations: (a) Pandemic experiences have affected students’ mental health and well-being adversely; and (b) specific factors such as isolation, concerns about physical health, academic concerns and more have driven students’ negative mental health experiences. Our study makes a significant contribution to the scholarship around COVID-19 and mental health of university students and has the potential to inform further research and action in this area. The in-depth understanding can be crucial during design of policies and support mechanisms by universities and higher education institutions, both in the UK and across the world, for improving students’ well-being and address their mental health needs in the aftermath of the pandemic.

## 2. Methods

### 2.1. Study Design and Setting

We conducted a qualitative study with undergraduate students at a public-funded university in the UK. We choose a qualitative design to explore the lived experiences of students during the pandemic which would in turn enable us to understand what is important to them, their felt needs and also provide a context to assess what further research and action is needed to improve students’ health and well-being.

The university is located in England and has an average enrolment of 19,000 students per year. The city where the university is located was one of the worst COVID-19-affected areas in the UK with high case and mortality rates and had to endure longer lockdown and restriction measures on account of this.

### 2.2. Population, Recruitment and Data Collection

The qualitative study was conducted in a follow-up of a questionnaire survey which was administered to 10,869 undergraduate students in June 2021 [35]. The questionnaire data were analysed and presented separately and only interview data are reported in this manuscript. A total of 827 students (7.6% response rate) completed the questionnaire and 187 of those expressed an interest to take part in the qualitative interviews. All 187 interested students were contacted at the email address provided by them, with an invitation and further information about the qualitative study. A total of 39 students responded to the invites and 34 students finally took part. Interviews were held between July and September 2021 and coincided with the COVID-19 vaccine roll-out to 18-years old in the UK. Data were collected by MG and LG through online interviews on Microsoft Teams and consent was secured verbally, which was recorded before the start of the interviews.

Participants’ demographic information were retrieved from their survey responses. Interviews lasted for an average of one hour and participating students were offered a £10 gift voucher as a token of appreciation. A piloted topic guide was used to conduct the interviews and probe questions were adapted if need was felt to explore emerging topics in the individual interviews (see Appendix A). Interviews were recorded with prior permission from the participants and transcribed either through Sonix, a transcription software, or manually by a professional transcriber. Transcripts were anonymised by the professional transcriber or by MG and LG for the machine generated transcripts.

### 2.3. Data Analysis

Data from the interviews were analysed inductively following the thematic analysis approach [36]. Both MG and LG started with reading the transcripts and interview notes to familiarise themselves with the data. Each researcher looked at transcripts of interviews done by the other researcher to make the process more inductive and minimise bias. Thereafter, MG started an open coding framework after coding the first five transcripts on NVivo which was shared with LG. Following this, the two researchers conducted joint coding exercises at online meetings and coded three additional transcripts on NVivo which helped refine the initial coding framework. The remaining transcripts were then independently coded by the two researchers using the framework, which was updated as new codes emerged. Initial themes were identified and rigorously discussed between MG and LG at regular meetings and agreed upon after final discussions with the wider team. The regular meetings also helped the researchers discuss any potential bias (e.g., LG as a university teacher or researchers’ own experience of the pandemic) that may have influenced the analysis.

## 3. Results

A total of 34 students participated in the interviews, out of which more than half (*n* = 21, 61.7%) were female. Details of participants’ characteristics are provided in Table 1.

The thematic analysis of students’ experiences brought to light two overarching themes and six sub-themes related to their mental health and well-being. The first theme presents students’ experiences of mental health and well-being during the pandemic. The second overarching theme deals with the factors that influenced students’ mental health and well-being. These include: (a) isolation; (b) concerns about physical health and well-being; (c) bereavement; (d) academic concerns; (e) financial worries; and (f) support, coping and resilience. These themes and sub-themes are explored in greater detail in the following paragraphs.

### 3.1. University Students’ Experiences of Mental Health and Well-Being during the Pandemic

A large number of participants (*n* = 26) shared about a range of negative emotions that they had experienced during the pandemic. Interestingly, most (*n* = 20) of these participants were female and proportionately they make up more than 95% of our sample of female participants. Several students, spoke about feeling ‘*frustrated*’, ‘*constricted*’ ‘*stressed*’ and/or ‘*depressed*’, particularly during the lockdown periods. Some of them went on to share how these emotions impacted their daily functioning, sometimes in considerable ways. In the words of one participant:


*I’m trying to think of a better word, but I would say it’s quite depressive. Honestly, because I’m quite bad when I get into a negative place, it’s quite hard for me to come out of it. And so I found that after quite a long time, you just sleep a lot. You know, you just watch TV and you don’t find much stimulus.*
(P32, Male, Social Science)

Another student spoke of her mental health needs during the pandemic:


*I feel like for me personally, like I feel like my mental health was very much impacted by the lockdown and the pandemic, because I’m someone who needs interaction with people to be happy, essentially. But I feel like this take away.*
(P2, Female, Humanities)

A few students also described how being at home and online learning blurred the boundaries between work and leisure leading to a lack of orientation of time and chaotic sleeping and eating patterns.


*So my timing is all changed. Like I had days I woke up and it’s like 4pm right now and it’s so-called lunch but it’s 4pm already. I have dinner at midnight and I don’t know what is going on but it is very unhealthy.*
(P1, Female, Natural Science)

For some, the restrictions gave rise to frustrations and anger, which in turn affected their relationships with their flatmates or family members that they were living with. As shared by one:


*We found there’s been more tension just from the fact that we’re all in such close proximity and you haven’t been able to just go out for the day, it’s made living quite difficult in a sense.*
(P26, Female, Life Science)

While there was a surge in the number of students’ seeking mental health support, the support services were often not at par to meeting such growing demand. Several of our participants who had sought either university or external mental health support found themselves facing long waiting lists and significant delays.


*Because I struggle with mental health conditions and also a lot of my friends and a lot of us were on the waiting list for support through wellbeing up until March or February and so many months have passed for us and the semester has passed without getting the support students need.*
(P33, Female, Life Science)

There were three participants in our interview cohort, who had a diagnosis of a mental health condition before the pandemic. Although it would be difficult to say if their mental health was worse as compared to the other participants, all of them shared about the negative emotions and feelings that affected their state of mental health during the pandemic.

### 3.2. Factors Influencing Students’ Mental Health and Well-Being

Several factors could be identified from the interviews, which had likely impacted students’ mental health adversely. Participants also spoke of factors which enabled them to cope or build their resilience in the face of adversity. Significant among these factors are the following.

#### 3.2.1. Isolation

The isolation brought on by social distancing and lockdown restrictions impacted almost all participants negatively and they described feeling ‘*sad*’ ‘*lonely*’ and/or ‘*anxious*’. While most students moved home (including international students) in March 2020 and never returned to campus, some kept switching between their term-time accommodation and home, depending on lockdown rules. Speaking of his experiences of staying in a student accommodation by himself, one participant shared:


*It felt very constricted and there were times where I felt like this flat was like a prison and I was going crazy. I was just pacing backwards and forwards. You run out of entertainment. You get sick of watching shows, you get sick of reading, it just starts to do your head in.*
(P31, Male, Social Science)

Reflecting on the challenge of social isolation, she experienced during the pandemic, another student said:

*I think there was a fear of isolation, I guess, especially when the first lockdown happened and we sort of were doing everything online, I think I was scared I was going to lose contact with my friends from uni*[versity]*. I think I was worried that I wasn’t going to make any new friends in seminars. I got quite down about that. I do remember that was sort of, yeah, at the beginning of the lockdown and sort of through the first lockdown, I remember feeling very worried that I was just going to lose the kind of social element of uni that I’d come to sort of rely on for a bit of socialising, a bit of social life. So yeah that was challenging, that was challenging, I remember.*(P13, Female, Humanities)

Several participants described how the isolation impacted their motivation to study or carry on with their other daily tasks.


*Staying at home, I really suffered from a lack of motivation. Like I’m really passionate about university and school, and I’ve always been one of those kids like who wants to learn more. But as soon after Christmas, my motivation just completely went. And I just found that I couldn’t be bothered to do anything. I would leave stuff like, not the last minute, but I would know it wouldn’t be my best work.*
(P2, Female, Humanities)

For some international students who had stayed back in the UK, the absence of family was greatly felt during this time. Alternately, some students who went back to their home countries, shared their anxieties about coming back to the UK as restrictions were being gradually lifted (despite high number of COVID-19 cases) at the time of conducting the interviews.


*I don’t really want to come back to UK. Well, I really want to stay in [home country] and continue remote study, because, you know, UK still got lots of COVID cases. And there’s the Delta!*
(P3, Female, Natural Science)

#### 3.2.2. Concerns about Physical Health and Well-Being

Concerns about their own well-being, or the well-being of their loved ones, also appeared to have had negative impacts on students’ mental health. Whilst many participants said that they considered themselves to be at low risk of being severely ill with COVID-19, there were a few who said that they were very worried about getting sick which made them fearful, overtly cautious and practise extreme avoidance. One of these participants said:


*I don’t leave the house, I don’t do anything…I don’t go anywhere. The only place I go is come back home and even then it would be my dad comes to pick me up, so from my front door to the car, which is about five steps and then it’s again from my driveway to my house so I wasn’t going anywhere.*
(P34, Female, Natural Science)

Another participant who said she was ‘*paranoid*’ of getting COVID-19, shared:


*I’m really afraid of getting it. Every time I go out, like every ten minutes I’m disinfecting my hands, sometimes I wear two masks and gloves. I’m a little paranoid about this! I believe if I caught it, my body would be strong enough to handle it, but I’m not sure, because there are a lot of variants of this disease, so I’m not sure if my body would handle it, but I’m really afraid of getting it. I almost don’t go out so I don’t get it.*
(P25, Female, Humanities)

Although personal risk was deemed to be low by most participants, the fear and anxiety about their family members being ill from COVID-19 affected many. The stress and anxiety seemed to have escalated when participants had family members fall ill with COVID-19. In the words of one participant:

*I have some relatives* [in Country X] *and at some point my whole family, they contracted COVID and two of them ended up in ICU for a short period and that was back in around November time and me and my parents were very anxious back then and worried and so, yeah, it’s just been taking a lot of mental health effects.*(P33, Female, Life Science)

Another student, whose parent was in hospital with COVID-19, described that period:

[Parent] *got it…but that was pretty traumatic because* [parent] *ended up going into intensive care…it was a pretty horrid experience but I think it was just made worse by the fact that we couldn’t leave the house and we were going like stir crazy at home worrying.*(P10, Female, Life Science)

Results also indicate that concerns about well-being were not limited simply to the virus but for some ethnic minority students, the fear of harassment or risk of facing hate crime also put a weight on their minds. This fear was grounded in their past experiences of racism which was magnified by reports of Asian communities being subjected to harassment because of the disease’s origin in China. As one participant shared:


*Well, to be honest, I haven’t been out that much in the pandemic…because of social media I have been worried to go out from what other people have told me or shared, people, especially being Asian, people call them Coronavirus or blame them for COVID and so it has worried me to go out because I would be blamed or be shouted at. And I do get anxiety about that.*
(P33, Female, Life Sciences)

#### 3.2.3. Bereavement

Three participants had lost a close relative, due to COVID-19 or other reasons, during the pandemic, which had an impact on their emotional well-being. Aside from the personal loss, the inability to mourn with family and not being able to conduct a proper funeral during the pandemic seemed to have made the bereavement process more difficult. As one participant whose grandparent had passed away narrated:


*It was weird as well, because obviously you can’t do your funeral like how it normally does because everyone’s isolating, you can’t see the body and things like that, so it was hard to process it afterwards.*
(P4, Female, Life Science)

Another participant who lost a close relative to COVID-19 shared how the death affected her. In her own words:

*I think emotionally, the first few days was hard because it was quick…I’ve had relatives die of things like cancer and you can prepare for that. You’ve got time in advance to prepare for that but there was no preparation and* [relative] *was* [young].(P8, Female, Natural Science)

#### 3.2.4. Academic Concerns

For quite a few of the participants, the concerns around the quality of education that they were receiving online and thoughts about future career prospects seemingly amplified their anxieties and worries. While a handful of students appreciated the flexibility that came with online learning, a larger proportion of participants said that classroom interactions between teachers and students, which are a big part of face-to-face teaching and learning, were lacking in the virtual classrooms.


*It’s definitely been my worst year of university so far…academically, I definitely feel as if I’ve learned less in this year than I have in the previous two years. I think that the content that’s put online, even though it does teach you the basic stuff, I don’t think it’s as engaging or I don’t feel like I’m learning as much as I would have learned if I was still attending in person lectures.*
(P19, Female, Life Science)

Some students also felt that their grades were affected because of these limited online interactions, which caused concerns about future aspects.


*You’re not having the same quality of support if you need help, you can’t like just go up to someone and ask for help, so it affected quite a lot of our grades as well…*
(P4, Female, Life Science)

Learning from home was also difficult for some participants who had caring responsibilities or had other family members working or schooling online at the same time. One mature student who was studying and home-schooling her children, described:


*Home schooling while studying myself; it was the worst part…To be honest, I used to get really mad and, like, it’s not a good thing, but I was like even like why don’t you understand like when I’m teaching, because I have to sit from let’s say nine o’clock in the morning till three/four…So there was a time when I really used to get really anxious and really angry.*
(P11, Female, Life Science)

Some students also spoke about deferring their exams or assignment submission or their studies because they found it difficult to keep up with the academic work while studying at home. But these decisions were not easy and fraught, both regarding the process and also about any repercussions. As one students shared:

*I don’t think universities understand how stressful it is to make a extenuating circumstance request* (A formal process initiated by most UK universities during COVID-19 pandemic to support students wherein a student can inform the university about any serious illness, bereavement or other significant event which might affect the student’s performance at assessment.)*…it feels awful to write a request…explaining why you couldn’t do an assignment…it’s just very draining…and also in terms of extenuating circumstance, if you get an extension it becomes like a domino effect, you know, like people who, if you ask extensions after extensions you end up falling behind.*(P33, Female, Life Science)

#### 3.2.5. Financial Worries

Related to academic concerns, another factor which apparently contributed to the stress experienced by some students’ was the financial aspect. These students expressed their worries about the education loans they had taken out and repaying these loans. Some also expressed pessimism that the online education had not equipped them with adequate skills for better future prospects. Some students also expressed dissatisfaction and frustration at having to pay tuition and accommodation fees at full-rates which they felt was not justified during the pandemic when teaching was online and accommodation was sparingly used.


*It is quite frustrating when you’re paying as much as you are to not have the full exposure to your degree and your course.*
(P32, Male, Humanities)


*I didn’t get the full university experience this year and that’s what really bothered me, I felt just lonely and almost like my money had been wasted because I’ve taken out a student loan and I have to pay this back and I can understand why the university switched to online learning but I feel like it was just Open University at this point and maybe we should have a refund or something.*
(P19, Female, Life Science)


*One of the worst things was that as a lab work based student…we finished first year without having the skills that we needed as we are moving to second year.*
(P15, Female, Life Science)

Speaking of the impact of financial concerns on their educational experience another student shared:


*I definitely felt like giving up, definitely, like, not just even in terms of education, like, I just, yeah, there was a lot of moments where I was like why, this has been such a waste of money, such a waste of years, I regretted taking a degree and trying to get a degree very much.*
(P4, Female, Life Science)

#### 3.2.6. Support, Coping and Resilience

While there were several stressors which likely impacted students’ mental health adversely during the pandemic, participants also spoke of the support they had, ways they coped and built their resilience during these difficult times. Most students spoke about the support they received from family and friends to help them cope with the pandemic.


*I mean my family have always been very supportive, we all sort of supported each other. It was all very you know let’s just stick together on this thing and it will all be OK so that was very nice.*
(P13, Female, Humanities)


*Like our friends came together a lot…we did that a lot online, checking up on each other, things like that.*
(P4, Female, Life Science)

Some students also appreciated the support they received from their personal tutors at the university.


*I’m quite close with my personal tutor…he’d email me…actively making sure I was OK and, like, I would email him any time I had questions about, like, what I needed to do in terms of mitigating circumstances, that kind of thing, making sure it’s OK for my mental health and my health.*
(P34, Female, Natural Science)

However, this appreciation was not universally shared and there were students who were unhappy with the lack of empathy and support from their teachers which they said made the pandemic experiences more difficult.


*Whenever I would reach out for support from my course, my personal tutor in particular, I would sort of be brushed off. I would just be given a pre typed message, which was exactly zero help. So I guess I was left quite frustrated because I wasn’t seeing them in person. And whenever I reached out, I wasn’t getting any support.*
(P28, Female, Medicine & Allied)

For some of our participants who were working part-time in frontline jobs or as clinical trainees, the ability to go out to their workplace, particularly during lockdowns, was something which they were thankful for and believed had helped them to cope. In the words of one participant:


*I still went out to work and work for me, like during the pandemic, work was my socialising. So I was actually one of the lucky ones that could go out and sort of socialise.*
(P10, Female, Life Science)

Some other students spoke of taking up new hobbies or exercising to keep themselves occupied.


*I also found that I really liked baking and I just kind of focused on baking and had that as a hobby and really tried to develop that as much as I could because I found that brought me a lot of joy.*
(P18, Female, Social Science)

Several other students also mentioned keeping a positive outlook and thinking positively to adapt to the changes brought in by COVID-19


*I think there was a moment of just sort of acceptance of like this is how it is now, this is the way it has to be for a while... I don’t want to complain about it, I don’t feel bitter about it.*
(P13, Female, Humanities)

## 4. Discussion

This is a qualitative study exploring students’ pandemic experiences and the impact of these experiences on their mental health and well-being. Although our study was not designed as diagnostic research, the experiences of mental health challenges shared by our participants concur with the findings of high levels of anxiety and depression among university students during the pandemic in other large-scale surveys, both in the UK and abroad [8,9,13]. Our participants expressed feelings of sadness, loneliness and worry that some found overwhelming which affected their daily functioning. Students shared how not being able to attend classroom lectures, meet family and friends, being in their rooms/homes at all times and/or fear of the virus affected their routine, their motivation and also their relationships. Disturbance in sleeping and eating patterns have also been mentioned by some participants indicating traces of poor mental health. While some of the outcomes should be viewed with caution as increased levels of anxiety may have been a temporary adaptation to the pandemic [29], however we note that although our study was conducted more than a year after the first lockdown, students’ anxieties were still persistent. Past studies on trajectories of mental health during epidemics have reported that while certain conditions like acute stress may decrease over time, but conditions such as post-traumatic stress disorder (PTSD), anxiety and depression may see a rise with passage of time [37,38]. Our findings are in line with the analysis of data from the Office for National Statistics (ONS), UK survey conducted in early 2021, which found that almost two-thirds of students reported a fall in their mental health and well-being since the start of the autumn term in 2020 [39].

The large proportion of female students in our cohort reporting negative mental health experiences is consistent with findings from other studies conducted before and during the pandemic [7,10,12,15,17,18,26,40,41,42]. The differences in the way stress and anxiety are experienced and reported by males and females could explain this gender imbalance [41,42]. This finding also indicates that while targeted mental health support for female students is necessary, awareness about mental health and reporting should be encouraged among male students.

Social isolation brought in by lockdown restrictions had been one of the most commonly cited reason by our participants for feeling negative during the pandemic. The impact appeared to have been more significant on those living in students’ accommodation or living by themselves as they experienced greater loneliness. This finding raises concern as previous studies, both before and after the pandemic, have evidenced the impact of loneliness on young people’s mental health, with greater loneliness associated with greater mental health problems [43,44,45]. Studies looking at social connectedness, loneliness and students’ mental health in the aftermath of the pandemic are needed to arrive at a clearer picture of any long-lasting impacts that the pandemic may have had on students’ mental health.

Starting university is a major life-event and marks the transition from school life to adulthood and higher education. It has been shown that the ability to make new friends and establish effective social interaction with peers help students adapt better to the transitions and demands of university life and also contribute to improved learning [46,47]. However, the reduced social interaction among students during the pandemic, and the anonymity in online classrooms, may have affected students’ ability to make friends, amplified their loneliness and negatively impacted their mental health and well-being [12,48]. Most participants in our study had described how they missed building friendships with their peers and classmates because they were unable to meet in person. Some of our first year students who were looking forward to their university life said that they were left disappointed by their experience of online learning and not being able to be a part of campus life. With students returning in large number to campuses, there are hopes that most would go back to leading a normal social life, but there might be a few who might struggle to navigate the social world after these forced periods of social isolation [49]. Further research is called for to understand what other mental health challenges await as pre-pandemic ways of university life return while the risk of the pandemic still continues.

Fear and anxiety arising out of anticipation of racist attacks or discrimination has also been reported by our ethnic minority participants. The rise in sinophobic hate crimes and xenophobia during the early phases of the pandemic [50] had created conditions where ethnic minority and international students have felt threatened about their safety and well-being, which in turn is likely to impact their mental health [51]. Oh et al.’s analysis of anxiety and depression among US college students have established a link between ethnic/racial discrimination and poor mental health [27]. Trammel et al. also found in their study among American undergraduates that Latinax and Asian students experienced higher COVID-19 related threat and negative beliefs than White students [52]. It is to be noted that ethnic minority students have been experiencing disproportionate mental health problems even before the pandemic due to inequality in access, gaps in academic attainment, unequal representation etc., and there is a strong likelihood that these factors may have exacerbated during the pandemic [53]. Universities should be more vigilant in identifying these at-risk groups and become proactive in dealing with instances of discrimination and, racially motivated hate crimes both inside and outside the campus. International students (some of whom are from ethnic minorities) also need to be supported by their institutions in raising concerns about racial discrimination and inform them of services that are available to support victims of discrimination and hate crime.

Our findings of negative impacts of academic and financial worries on students’ mental health have been reported by other studies [8,9,54]. Although academic concerns impacting students’ mental health is not new [40], but pedagogic challenges unique to the pandemic (e.g., missing out on practical lessons and/or field excursions, limited interaction with teachers during online classes and, online assessments) have been mentioned by our participants as having added to their anxiety and stress. While support mechanisms had been initiated by universities to provide academic support to students, but as mentioned by some of our participants, some of the processes itself had been quite fraught. These experiences need attention as poor mental health has been found to be associated with higher drop-out rates among students in higher education, which can have far-reaching adverse consequences on students’ later lives [55].

Expenditures incurred by university students for their education have also been known to have an effect on their mental health [56], but this may have intensified during the pandemic raising concerns about students falling into the ‘vicious cycle’ where financial difficulties beget mental health problems which in turn exacerbate students’ financial difficulties [57]. Amongst our participants too, concerns about repayment of education loans, fees, accommodation costs made some students question their decision to enrol in university during the pandemic which seemed to have impacted their mental well-being.

Our finding on support, coping and resilience during the pandemic explains how most students have managed to overcome the mental health challenges brought in by the pandemic. Previous studies have shown how optimism and positive coping (which in turn are related to higher social support) among students can prevent mental health problems or vice-versa [9,58,59]. Furthermore, our research demonstrates how support from personal tutors in difficult times can help students cope, not only with the academic pressures, but also with the emotional burden and stresses that they face in their personal lives. Universities should look at institutionalising these support mechanisms going forward and also facilitate tutors in performing this role alongside their academic duties. This could also be helpful in light of the shortage of mental health support services as highlighted by our participants and also by other research studies [60,61]. Additionally, our data alludes mostly to adaptive coping strategies wherein students diverted their negative feelings by engaging in new hobbies or thinking positive thoughts. But there might be cases where maladaptive coping (e.g., drug or alcohol abuse, internet addiction) had been adopted by students to deal with the pandemic related stressors, which need further research and attention.

Our study has certain limitations. First, the study was conducted in the summer of 2021 and hence there is retrospective reporting about mental health challenges experienced by students, mostly during the lockdown periods. The findings, however, hold relevance in understanding how students perceived their mental health at a time when pandemic restrictions are gradually being eased and campuses are being opened. Second, our study was conducted in only one university and hence experiences may not be generally applicable. Nevertheless, the diversity of our participants in terms of ethnicity, course of study and residence status offers a wide lens for viewing students mental health needs.

## 5. Conclusions

In conclusion, our study presents rich qualitative data about university students’ experiences which influenced their mental health and well-being during the COVID-19 pandemic. One key outcome was identification of the major factors (isolation; health and, well-being; bereavement; academic concerns; financial worries and support, coping and resilience) that have influenced students’ mental health and well-being. Another key finding was the importance of support from family, friends and the institution for maintaining the mental health and well-being of students during an extremely stressful time. The findings indicate problems with mental health among students during the pandemic and call for further cross-sectional research to establish prevalence of mental health problems on different student groups (such as students who are female, belong to an ethnic minority community or international) in the aftermath of the pandemic. Our results also support the urgency for improving university mental health services as students are entering the higher education system with the burden of pandemic experiences from the past and adjusting to life in a post-pandemic world.

## Figures and Tables

**Table 1 ijerph-19-09322-t001:** Characteristics of interview participants.

Participants’ Characteristics	*n* (%)
Sex	
Female	21 (61.7%)
Male	13 (38.3%)
Ethnicity	
White (including non-British White)	19 (55.9%)
Asian	10 (29.4%)
Black	2 (5.9%)
Other	3 (8.8%)
Age	
≤22	20 (58.8%)
22+	14 (41.2%)
Year of study	
First (including Foundation)	13 (38.3%)
Second	14 (41.2%)
Third	7 (20.5%)
Course	
Humanities	12 (35.2%)
Law	4 (11.8%)
Life Science	8 (23.5%)
Medicine & allied	4 (11.8%)
Natural Science	4 (11.8%)
Social Science	2 (5.9%)
Residence status	
UK student	26 (76.4%)
UK-based international students	4 (11.8%)
Non-UK based international students	4 (11.8%)

## Data Availability

The data presented in this study are available upon reasonable request from the corresponding author. The data are not publicly available due to ethical reasons.

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
