# Peer review of "University Students’ Mental Health and Well-Being during the COVID-19 Pandemic: Findings from the UniCoVac Qualitative Study"

_ijerph, 2022, doi:10.3390/ijerph19159322_

Round 1
Reviewer 1 Report
Dear Authors,
thanks a lot for your extensive additions.
I see all my comments sufficienty addressed.
Author Response
We thank the Reviewer for taking time out to read through our revised submission and appreciating the changes we have made to our manuscript following the comments.
Reviewer 2 Report
I agree with the authors' revision.
It's better to emphasize why you should choice the method of interview (e.g., why not a quantitative method) and what is the theoretical contributions.
Good luck.
Author Response
We appreciate the Reviewer’s suggestion and have revisited our manuscript in the light of this comment. We have noted the rationale for undertaking a qualitative study in our Introduction (Lines 140-152) section and the contributions of our research has been made throughout the Introduction and Discussion sections. However, we do take the Reviewer’s comment into consideration and have revised sub-section 2.1. Study Design and Setting within the Methods section to explicitly state the choice of methodology. The new sub-section (Lines 155-168) reads as below:
We conducted a qualitative study with undergraduate students at a public-funded university in the UK. We choose a qualitative design to explore the lived experiences of students during the pandemic which would in turn enable us to understand what is important to them, their felt needs and also provide a context to assess what further research and action is needed to improve students’ health and well-being.
The University is located in the England and has an average enrolment of 19,000 students per year. The city where the University is located was one of the worst COVID-19 affected areas in the UK with high case and mortality rates and had to endure longer lockdown and restriction measures on account of this.
This manuscript is a resubmission of an earlier submission. The following is a list of the peer review reports and author responses from that submission.
Round 1
Reviewer 1 Report
It is my pleasure to review the manuscript titled “University Students’ Mental Health and Well-Being during the 2 COVID-19 Pandemic: Findings from the UniCoVac Qualitative Study” submitted to “Int. J. Environ. Res. Public Health” for consideration to be published.
This study draws on a sample of 34 undergraduate students, conducted indepth interviews to understand the influence of COVID 19 on students’ mental health and well-being. The interviews were a follow up to an larger questionnaire.
The authors assess a timely and important topic which is highly relevant to higher education providers in UK and abroad and even more so given that some nations have not given up a zero-covid policy.
My comments follow the structure of the paper
Abstract.
The abstract is well written outlining the importance of the study, the method and findings. To make the paper more relevant I would have positioned the paper within the literature and express the contributions (mainly theoretical)
Introduction
The introduction is brief and could benefit from additional literature. A short outline of findings and contributions is necessary to attract readers attention.
To me the introduction lacks a clear positioning on who is the target audience and how to provide them with relevant finding and contributions. As of no it is rather ‘universially’ written.
A section on Literature review is missing
2. Methods
The selection of interview participants is not clear. It triggers the issue of selection bias and or non-response bias for students who were invited to be interviewed but did not turn up.
3.Results
This section is purely descriptive not link to expectations (based on the prior questionnaire) not links to existing literature.
4. Discussion
This section is shallow and superficial much more efforts are needed to convince readers about the validity of your finding to be linked to existing literature. Again, I found it difficult to see contributions within this section.
The limitations are well stated, however they also cause high doubt on the relatability and validity of this study.
5. Conclusions
This section is too short and again lacks contributions to practice and theory.
References
More references to Int. J. Environ. Res. Public Health are needed to show the relevance of your study to this journal
Reviewer 2 Report
The research topic is important. However, I believe that the manuscript must be improved. Especially, the authors must clarify the originality and theoretical contributions of the study.
To clarify the originality and theoretical contributions, you should show research (analysis) angle through theoretical framework foundation. Otherwise, for example, what is the section 3.1 and how does it relates to discussion is unclear. Another example is the section 3.2.2. It is unclear how did you categorize "Health", "Well-Being", and "Bereavement" in the same category. The authors said "The second overarching theme deals with the factors that influenced students’ mental health and well-being" (lines 119-120), but its sub-category includes well-being again. What do you mean well-being factor influence well-being???
In the section 3.2.3, the authors should arrange the results (interview data) and clarify the differences with previous studies (i.e. features of the COVID-19 contexts).
In the section 3.2.5, it is unclear how this factor relates to well-being. In addition, the interview data told about support not coping and resilience. How did you give the code "coping and resilience"?
Which data and results relate to the paragraph of lines 351-357?
[Lines 380-390] This paragraph is unclear. If you want to discuss management, you should arrange the results again. For example, you can separate the results into "resource of coping and resilience" and "process of coping and resilience." Then, you can deepen the discussion. Now, the authors only show one or two examples. Therefore, it is difficult to discuss management of measures for the pandemic.